# Drug Repurposing and De Novo Drug Discovery of Protein Kinase Inhibitors as New Drugs against Schistosomiasis

**DOI:** 10.3390/molecules27041414

**Published:** 2022-02-19

**Authors:** Bernardo Pereira Moreira, Michael H. W. Weber, Simone Haeberlein, Annika S. Mokosch, Bernhard Spengler, Christoph G. Grevelding, Franco H. Falcone

**Affiliations:** 1Biomedical Research Center Seltersberg (BFS), Institute of Parasitology, Justus Liebig University Giessen, 35392 Giessen, Germany; bernardo.pereira-moreira@vetmed.uni-giessen.de (B.P.M.); michael.weber@vetmed.uni-giessen.de (M.H.W.W.); simone.haeberlein@vetmed.uni-giessen.de (S.H.); christoph.grevelding@vetmed.uni-giessen.de (C.G.G.); 2Institute of Inorganic and Analytical Chemistry, Justus Liebig University Giessen, 35392 Giessen, Germany; annika.s.mokosch@anorg.chemie.uni-giessen.de (A.S.M.); bernhard.spengler@anorg.chemie.uni-giessen.de (B.S.)

**Keywords:** schistosomiasis, protein kinase inhibitors, drug repurposing, de novo drug repurposing, AP-SMALDI MSI

## Abstract

Schistosomiasis is a neglected tropical disease affecting more than 200 million people worldwide. Chemotherapy relies on one single drug, praziquantel, which is safe but ineffective at killing larval stages of this parasite. Furthermore, concerns have been expressed about the rise in resistance against this drug. In the absence of an antischistosomal vaccine, it is, therefore, necessary to develop new drugs against the different species of schistosomes. Protein kinases are important molecules involved in key cellular processes such as signaling, growth, and differentiation. The kinome of schistosomes has been studied and the suitability of schistosomal protein kinases as targets demonstrated by RNA interference studies. Although protein kinase inhibitors are mostly used in cancer therapy, e.g., for the treatment of chronic myeloid leukemia or melanoma, they are now being increasingly explored for the treatment of non-oncological conditions, including schistosomiasis. Here, we discuss the various approaches including screening of natural and synthetic compounds, de novo drug development, and drug repurposing in the context of the search for protein kinase inhibitors against schistosomiasis. We discuss the status quo of the development of kinase inhibitors against schistosomal serine/threonine kinases such as polo-like kinases (PLKs) and mitogen-activated protein kinases (MAP kinases), as well as protein tyrosine kinases (PTKs).

## 1. Introduction

Schistosomiasis is an infectious disease that affects millions of people worldwide, and besides many preventive measures, the treatment relies mainly on one drug—praziquantel (PZQ) [1,2]. PZQ is highly effective as a single dose against all *Schistosoma* species, has few adverse effects, and is cheap to produce [3]. However, PZQ is not able to kill eggs or juvenile worms, limiting the drug efficacy against the parasite, and does not protect against reinfection [4,5]. Furthermore, since PZQ is the only commonly used drug against schistosomiasis, there is justified fear of emerging resistance [6]. Hence, there is still an unmet need to find new antischistosomal drugs, ideally ones engaging the aforementioned issues. While no vaccine against schistosomiasis has been completely developed [7], the search for alternative chemotherapy for this disease, however, relies on the laborious and usually expensive process of drug discovery.

As seen for cancer, autoimmune, and many other diseases, protein kinases (PKs) are in the spotlight when it comes to target identification for drug therapy. PKs are enzymes that selectively modify substrates by adding a phosphate group in a process called phosphorylation. The resulting modified target will then change its functional activity, such as localization, association to other molecules, and status of enzymatic activity (e.g., in signaling cascades) [8]. As PKs greatly affect cell biology, their activity is highly regulated, and any loss of this fine control can lead to abnormal cell behavior, and in some cases, to disease development, such as cancer [9]. Accordingly, due to tremendous effort put into the study of human kinases as molecular targets, there are now many successful kinase inhibitors approved for use. From 52 FDA-approved drugs against kinase proteins, at least 21 are considered multikinase inhibitors [10]. Moreover, the PKIDB (https://www.icoa.fr/pkidb/index.html (accessed on 20th December 2021)), an updated and curated database of PK inhibitors, lists more than 220 different compounds that are being tested in ongoing clinical trials worldwide against several diseases [11,12].

The approaches that have been applied for drug discovery against schistosomiasis are encompassed in two main categories—namely, de novo drug development and drug repurposing. The classical pathway in drug discovery, or de novo drug development, usually starts with the identification of targets of interest. Molecular and genetic tools help characterize the functional role of the targets, and researchers can select those that present potential druggability to perform numerous assays. Then, only after laborious screening and careful identification of many compounds, lead candidates can be optimized and developed for clinical trials before they are finally approved for market use [13]. This path can take anything up to 20 years and is very risky, considering that any unexpected negative result at any stage can bring the entire drug development process to a halt. Traditional de novo drug discovery has a low overall probability of success (<1% of promising lead molecules will be tested in humans), though the return for any financial investment can be relatively high (Table 1) [13]. Usually, the pharmaceutical business tends to favor cost-effective and profitable discovery areas. For neglected tropical diseases (NTDs), however, there is no assurance that the economic returns will be substantial [14].

This underlines drug repurposing, or repositioning, as a more efficient and cost-effective alternative, compared with the abovementioned traditional approach. Drug repurposing aims at applying an existing drug to a different disease as means of developing new therapeutics for diseases without optimal treatment or any treatment at all [15]. These methods present reduced drug development time, and fewer risks regarding safety and pharmacokinetic properties, which translates to a higher probability of success, something that the industry and money investors tend to support. Regarding NTDs, such as schistosomiasis, where market funding is generally low, drug repositioning is of great interest [16].

The recent effort in the search for new kinase-targeting compounds against schistosomiasis has been put forward mostly by drug repurposing approaches. The plethora of kinase inhibitors that have been identified and characterized in research for cancer, autoimmune diseases, and complex syndromes makes drug repurposing a much more appealing path. Nevertheless, we are still to find a suitable alternative to PZQ, especially concerning PK inhibitors (PKIs). Here, we highlight recent studies that focused on the discovery of new kinase inhibitors against schistosomiasis, encompassing the two aforementioned approaches.

## 2. Discovery of Natural Compounds as Source for Anthelmintic Drugs

One vast source for novel and optimizable molecules for de novo drug discovery and development is natural compounds. Historically, natural products and their derivatives have been the source for the majority of developed and approved drugs [17]. Natural products have many advantages such as providing complex molecules with considerable structural variability. However, the advantages over synthetic compounds stop there, since access, supply, and purity of natural molecules can sometimes be a limitation [17]. Several natural compounds have been shown to be active against schistosomes [18] and other parasites [19]. For instance, the diarylheptanoid curcumin showed a schistosomicidal effect in vitro against adult worms from two species (*S. mansoni* and *S. haematobium*), with effective inhibition of egg production [20,21]. It was suggested that this activity could be due to the action of curcumin on the ubiquitin–proteasome pathway, although the mechanism was not clear [18]. Another finding that has also been reported is that curcumin and its many derivatives have potent anticancer activities by targeting protein tyrosine kinases (PTKs) [22]. Moreover, a recent study elucidated the mechanism in which curcumin targets the dual-specificity tyrosine-regulated kinase 2 (DYRK2) to reduce proteasome activity [23]. Whether curcumin can also target *Schistosoma* PKs and exert schistosomicidal effects by inhibiting proteasome activity remains to be elucidated.

A similar situation applies to quercetin. This compound and some derivatives were isolated from natural plant sources in Brazil and tested in vitro for their schistosomicidal activity [24]. The quercetin-derived flavonoid quercitrin exhibited significant efficacy against adult worms, causing reduced motor activity up to 87.5% and the death of 25% of both male and female worms. Although the mechanism by which quercetin affects schistosomes is still unclear, many studies have characterized the properties of quercetins as multikinase inhibitors [25]. However, it is not known if this activity is also observed against *Schistosoma* PKs.

Another isoflavonoid-derived compound, genistein, is a characterized PTK inhibitor, among other pharmaceutical properties [26,27]. It was discovered in the root-tuber peel extract of the leguminous plant *Felmingia vestita*, a common food source in South Asian communities, and used as a traditional anthelmintic by the Khasi tribes of India [28]. It has also been investigated against several commonly occurring helminth parasites, among them the tapeworms *Echinococcus multilocularis* and *E. granulosus*, *Fasciola hepatica*, and *Ascaris suum* [27]. It is known that some PTK inhibitors have potent activity against hepatic fibrosis, which is usually a common outcome of chronic schistosomiasis [29]. One study used genistein to treat *S. mansoni*-infected mice, and the data showed some positive results such as reduction in worm burden, egg load, and fewer and smaller liver granuloma formation when used in combination with PZQ [30]. Due to the broad activity genistein has against several parasites, and considering the strong effects of flavonoids in cancer [31], additional studies would be necessary in order to explore new possible derivatives from this class of compounds against key PK targets from schistosomes.

Some other studies have investigated the inhibitory activity of many natural molecules against PKs, with some of them reported functioning as kinase inhibitors [25,32]. For instance, artemisinins are well-known antimalarial drugs, and their use as schistosomicidal drugs has been already explored. Although these drugs were shown to target the phosphatidylinositol-3-kinase (PI3K) in malaria parasites [33], no investigation has been made regarding the mechanism in schistosomes. On the other hand, many natural compounds are demonstrated to exert cellular effects on PK signaling cascades; however, no direct target–compound interaction has been shown. This is the case for vernodalin, which has been also shown to have antischistosomal effects [34], and a recent publication addressed the close relationship of this class of compound to the activation of MAP kinase signaling pathway with some anticancer effect [35]. Similarly, piplartine has been reported to have antischistosomal effects [36,37,38], and an extensive review has already been published covering its many effects, including alterations in PK signaling cascades [39].

An alternative to the extensive search for natural compounds is the development of synthetic compounds from promising lead natural scaffolds. Nowadays, the pharmaceutical industry is capable of creating numerous compounds through synthetic processes, i.e., making novel molecules from available natural precursors, which contributes greatly to the drug design field [40,41]. A synthetic substance refers to a molecule that can be created, rather than being produced by nature. It also refers to a compound formed under controlled chemical reactions, either by chemical synthesis or by biosynthesis. Given the current state of knowledge, synthetic methods can deliver biologically active and structurally intricate molecules in a cost-effective manner. Accordingly, this area has also been broadly addressed for schistosomiasis, in which several different classes of synthetic compounds have been created, optimized, and studied, such as biaryl alkyl carboxylic acids [42,43], trioxolanes [44,45,46], aryl ozonides [47], tetraazamacrocyclics [48], etc. [49,50,51,52,53,54,55,56]. Although most of these synthetic derivatives show satisfactory antiparasitic properties against Schistosoma in vitro and/or in vivo, there is still a wide knowledge gap regarding ligand–target interactions. Since the studies have been focused on the effects and toxicity of such compounds, little or no attention has been given to the identification of the actual targets from these synthetic inhibitors. Certainly, adding new information regarding antischistosomal synthetic compounds, and PKs would be valuable for the development of new treatments for this disease.

It is clear that possibilities for exploring inhibitory effects of natural and synthetic compounds against schistosome kinases exist; however, there is still a need for more evidence of compound–target interaction in this organism. Further down the path of drug discovery of natural compounds, strong investment in drug development and optimization is required after the newly identified compounds have shown promising properties and activity. Negative aspects of natural compounds, such as low stability and solubility, can usually be improved by computer-aided drug development approaches. Nevertheless, finding exactly which protein targets are inhibited by these new molecules in *Schistosoma* is a time-consuming and challenging task. That is where drug repurposing emerges as an alternative to accelerate drug discovery.

## 3. Protein Kinases as Central Targets for Drug Repurposing

The current state of available technologies to develop drugs has led to an expansion of novel therapeutics for many diseases. Likewise, there was a substantial expansion of many libraries and collections of compounds that could be easily supplied and explored for other disease models. In addition, specific collections allowed researchers to focus on compounds targeted against one or more diseases (e.g., Pathogen Box—https://www.mmv.org/ (accessed on 20th December 2021), NCI Oncology Set—https://dtp.cancer.gov/ (accessed on 20th December 2021), NIH Clinical Collection, etc.), or to investigate drug–target interactions, as is the case for PKs (PKIDB [11,12], Selleckchem Kinase Inhibitor Library, Enzo Kinase Inhibitor Library, etc. [57]). Though the Pathogen Box is a diverse compound collection, it does contain many kinase inhibitors that could potentially work against schistosomiasis [58]. The collection is made up of 400 compounds grouped into several disease sets according to their potential against the most important NTDs, including schistosomiasis. A multicenter screen using this collection and combined strategies identified 35 core compounds that could be used in further investigations [59]. From those, 13 were considered promising leads, with 10 of them originating from disease sets other than schistosomiasis (Figure 1). This reinforces how the investment in drug repurposing could boost the drug discovery process.

Moreover, given the conservative nature of PKs across organisms, repurposing approved anticancer drugs, which mostly target PKs, is also a valid alternative. Cowan and Keiser explored the latter approach by screening 114 compounds from the Developmental Therapeutics Program (DTP) of the National Cancer Institute, USA, against larval and adult stages of *S. mansoni*. A total of 11 compounds presented satisfactory IC50 values (<10 μM), with 2 kinase inhibitors—trametinib and vandetanib—showing positive in vivo results (Figure 2A) [61].

In the search for new ways to treat schistosomiasis, researchers are exploring drug repurposing as an alternative method to test drugs for combined use with PZQ. Investing in combined therapy alongside PZQ is a valuable alternative to overcome the limitations of the treatment with PZQ alone. The aim of this strategy is to achieve additional or synergistic therapeutic effects promoting the minimal or delayed appearance of drug resistance [62]. Many drugs and derivatives have been extensively reviewed, but so far, not many have stood out as new chemotherapy against schistosomiasis [62,63,64,65]. With respect to PK inhibitors combined with PZQ therapy, in 2013, You et al. identified some key regulatory genes that are upregulated in response to PZQ treatment, among which was the Ca^2+^/calmodulin-dependent PK II (CaMKII). The authors found that inhibition of CaMKII led to improved efficacy of PZQ treatment [66]. In a more recent study, the authors tested a range of commercially available CaMKII inhibitors (selective and non-selective) in PZQ-sensitive and -insensitive schistosomula [67]. Several inhibitors presented good in vitro efficacy, suggesting that the inhibition of CaMKII is a valid alternative as an adjunct therapy to PZQ in treating juvenile and adult schistosome infections. Two drugs—staurosporine and 1-Naphthyl-PP1—presented promising efficacy when used in combination with PZQ, and further investigation would be necessary to improve their efficacy (Figure 2B) [67]. Although staurosporine is known as a universal kinase inhibitor and has been widely used in research, its non-selectivity and increased toxicity as a drug hindered some of its pharmacological use [68,69], explaining the investigation of such molecules in combinatorial drug therapy approaches.

As important as finding compounds with drug-like properties and satisfactory efficacy in in vitro assays is, it is also crucial to select the best suitable targets for drug therapy and development. Some studies have taken advantage of exploring the relationship between human and schistosome kinases. For instance, *S. mansoni* has two polo-like kinases (PLKs), in contrast to the five PLKs found in humans. PLKs are serine/threonine kinases tightly involved in cell division processes, and in schistosomes, PLKs are expressed in sporocysts and in the gonads of adult worms [70]. Following their functional characterization by RNAi, Long et al. screened 11 known human PLK inhibitors to assess their bioactivity againt larval and adult worm stages of *S. mansoni* [71]. A phase I clinical candidate, a benzimidazole thiophene compound named GSK461364 (Figure 2C), showed high potency in the low micromolar range, serving as a reference for finding 38 other analogs by structure–activity relationship (SAR). The in vitro results against schistosomula and adult worms were satisfactory, suggesting that PLKs are good druggable targets in this parasite.

As a consequence of the increased availability of the genomes of different *Schistosoma* species, the kinome was also further characterized [72,73,74,75]. This allowed researchers to predict with more accuracy which PK inhibitors approved for human use could possibly present efficacy for targeting schistosome kinases, as performed for *S. haematobium* [73]. A bioinformatics study that helped elucidate the kinome of *S.*
*japonicum*, based on human kinase drug–target interactions, was able to screen 97 FDA-approved drugs predicted to target 157 kinase targets from *Schistosoma*. Among the top hits retrieved from the phenotypic screening were tyrosine kinase inhibitors (TKIs), such as imatinib (Figure 2D), and some second-line TKIs such as bosutinib, crizotinib, nilotinib, and dasatinib [74]. These observations suggest that computational approaches for drug repurposing can work as starting points in the discovery of new schistosomicidal agents.

PTKs are very important components of cellular processes such as development and metabolism and are highly targeted in cancer therapy, as a means to stop cell proliferation [76]. The so-called tyrphostins (tyrosine phosphorylation inhibitors) presented good efficacy in inhibiting PTKs, and many were approved for use in humans [76]. Consequently, this class of inhibitors was further investigated against schistosomiasis. Many studies have characterized several PTKs in schistosomes in recent times [77,78,79,80,81], which include PTK members of the insulin receptor, IGF, Src, Syk, and Abl subfamilies. Thorough reviews have already summarized the drug–target interactions in this field [82,83]. For instance, tyrphostins AG1024, AG538, and HNMPA-(AM)3 have potential effects on the survival of adult worms of some *Schistosoma* species (Figure 2E) [78,81].

Herbimycin A, a specific inhibitor of Src kinase, was able to reduce male-induced mitotic activity in paired females [84], and piceatannol, a known Syk kinase inhibitor, affected parasite gametogenesis [80]. Imatinib has been described as a potent drug against adult and juvenile *S. mansoni*, affecting morphology, pairing, and survival [85] (Figure 3). Beyond that, imatinib has also shown activity against *S. japonicum* [86], *Echinococcus multilocularis* [87], *Fasciola hepatica* [88], *Brugia malayi* [89], *Loa loa* [90], *Leishmania major* [91], *Giardia lamblia* [92], and it prevented erythrocyte egress of *Plasmodium falciparum* [93]. In a case report, a single 600 mg dose of imatinib was even shown to reduce the number of microfilariae in a patient infected with *L. loa* [94]. Analysis of the route of imatinib uptake, its tissue tropism, and metabolization in schistosomes was recently achieved by a technique called MALDI mass spectrometry imaging [95] (Figure 4) and can likely be adopted for the study of other PTK inhibitors. Altogether, this suggests that imatinib, its derivatives, or further compounds of the “*tinib*” PK-inhibitor class should be systematically explored for their antiparasitic potential [96].

Next to PTKs, the family of serine–threonine kinases is the focus of research. Among these, mitogen-activated PKs (MAP kinases), such as SmERK, SmJNK, or Smp38 in *S. mansoni*, have been shown to be important in development, fertility, and protection against reactive oxygen species, as demonstrated by RNAi experiments [97,98]. Our laboratory is, therefore, investigating MAP kinases as novel targets. As no schistosomal PK structures are currently available, our research relies on using a combination of structural predictions (Figure 5) and biochemical approaches.

The lack of options regarding the treatment of human parasitic diseases, especially schistosomiasis, highlights the urgent need for new anthelmintics. The recent studies so far have strengthened the evidence showing that PKs are good druggable targets in schistosomes and further parasites, especially PTKs, which have been intensively investigated. Drug repurposing has been the most promising strategy in many studies focused on using *Schistosoma* PKs as molecular targets, and the success of using this class of inhibitors to treat cancer and chronic diseases might translate to the development of novel alternatives to treat parasitic diseases. The advances in parasite basic research and the new technologies for computer-aided drug design have widened the possibilities for target-based therapy, making it possible to fully explore the potential of PKs in the fight against schistosomiasis. Although several targets have been investigated, many promising drugs are mainly effective against adult worms, just as it is known from the PZQ mode of action. This would perhaps result in a “more of the same” drug, limiting the development of an “alternative” treatment. Therefore, it must be paramount also to invest in new drugs that are active on juvenile worms as well.

## Figures and Tables

**Figure 1 molecules-27-01414-f001:**
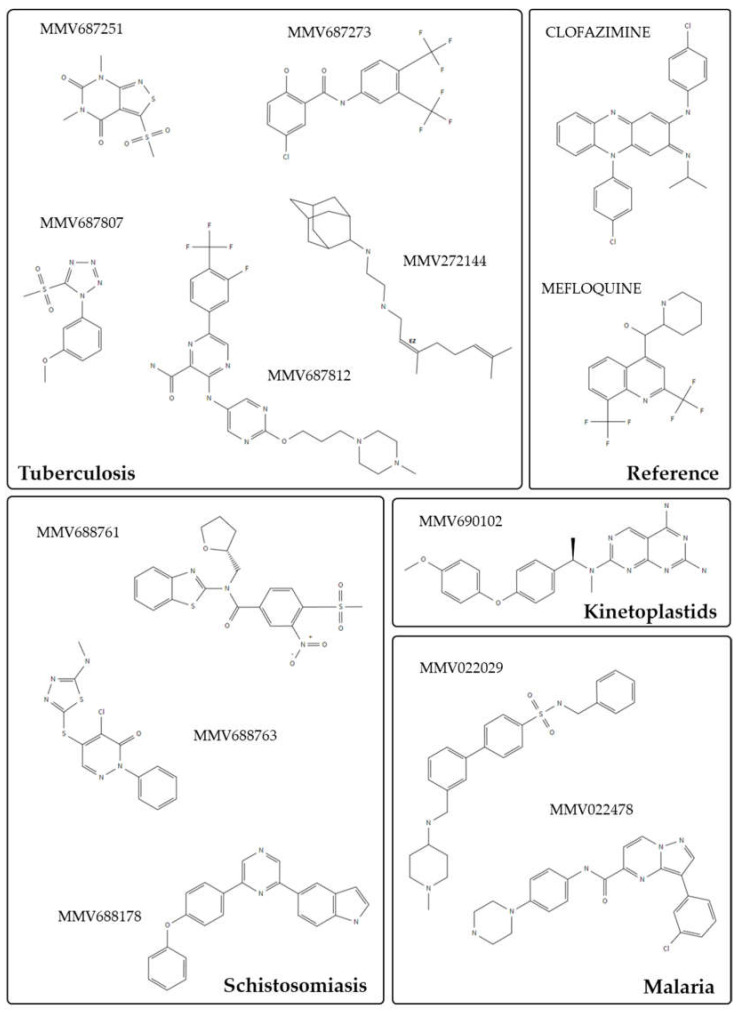
Two-dimensional (2D) structures of the most promising compounds from the Pathogen Box used in the repurposing study against schistosomiasis [59]. Of 13 compounds that were identified in this study, 10 belong to different disease sets other than schistosomiasis. Compounds grouped into the same disease or reference sets are represented by different boxes, and 2D structures were retrieved from PubChem database; CID: 89585418, 71596999, 71508634, 2221997, 22315759, 122196562, 44523667, 44524414, 2794, 4046, 40207755, 16605695, and 122196571 [60].

**Figure 2 molecules-27-01414-f002:**
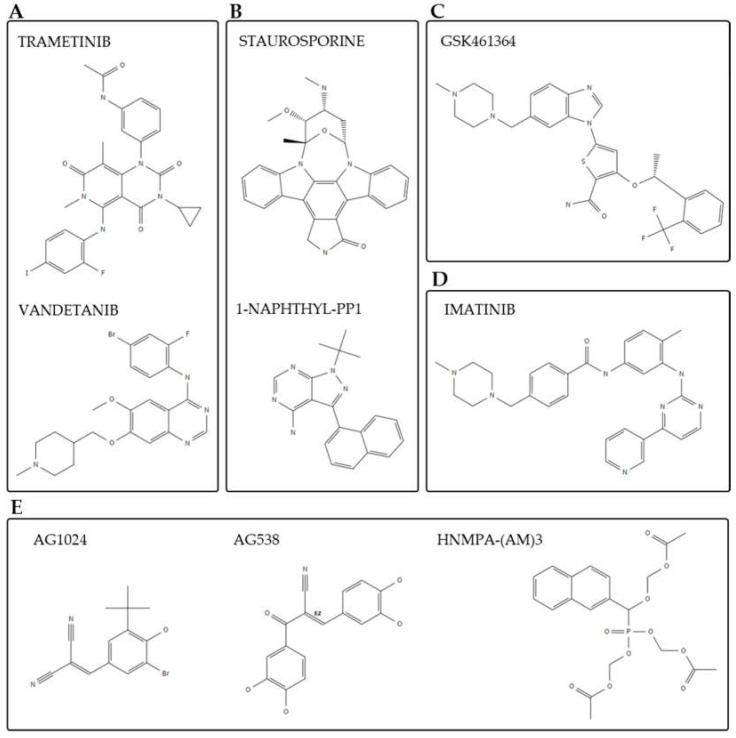
Two-dimensional (2D) structures of the most promising compounds used throughout several drug repurposing studies against schistosomiasis. Compounds are grouped according to the different studies: (**A**) [61], (**B**) [67], (**C**) [71], (**D**) [74], and (**E**) [78,81]. The 2D structures were retrieved from PubChem database; CID: 11707110, 3081361, 44259, 4877, 15983966, 5291, 2044, 5328760, and 3619 [60].

**Figure 3 molecules-27-01414-f003:**
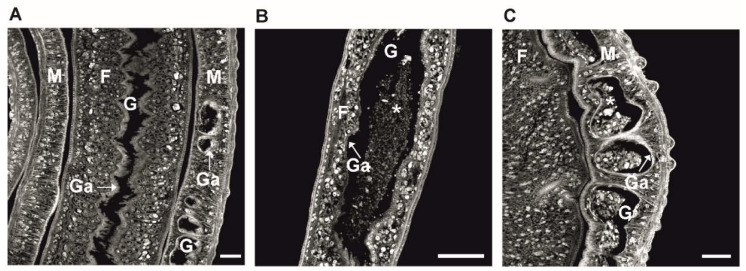
Confocal laser scanning microscopy of adult *S. mansoni* after treatment with imatinib in vitro: (**A**) an untreated control couple showing normal gut (G) morphology. In both sexes, the gut lumen is enclosed by a thick, syncytial gastrodermis (Ga). In paired schistosomes, the female (F) lodges inside the gynaecophoric canal of its male (M) partner. Thus, part of the male body encircles the female; (**B**,**C**) representative examples of *S. mansoni* female (**B**) and male (**C**) following treatment with imatinib (100 µM) in vitro for 24 h. One of the effects of imatinib is the degradation of the gastrodermis in both sexes, which leads to the accumulation of tissue residues inside the gut lumen (stars), and to the death of worms. Scale bars: 100 µm.

**Figure 4 molecules-27-01414-f004:**
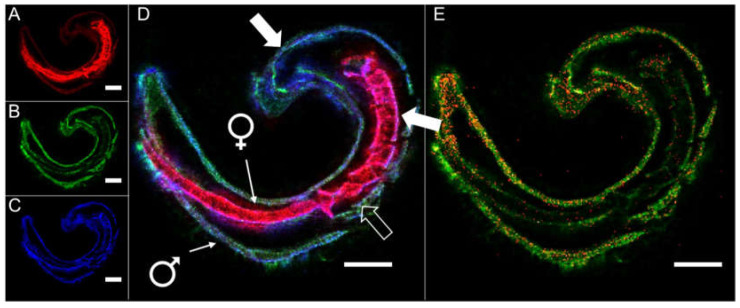
Uptake, tissue tropism, and metabolization of the Abl kinase inhibitor imatinib by *S. mansoni* revealed with atmospheric-pressure scanning microprobe MALDI mass spectrometry imaging (AP-SMALDI MSI). MSI images of a tissue section of an *S. mansoni* couple treated for 20 min with imatinib (100 µM): (**A**–**D**) imatinib is taken up by the male and female worm. Depicted analytes are *m*/*z* 579.534686, found to be enriched in the female (**A**, red, DG(34:0)), *m/z* 494.266284 representing imatinib (**B**, green) and *m*/*z* 760.585083, found to be enriched in the tegument (**C**, blue, PC(34:1)); (**D**) RGB image after merging images from (**A**–**C**). Imatinib accumulated amongst others in the tegument of both sexes. Filled arrows: tegument; unfilled arrows: intestine; (**E**) imatinib is metabolized to *N*-desmethyl imatinib in *S. mansoni*. Depicted are *m/z* 494.266284 representing imatinib (green), and *m/z* 480.250634 representing the metabolite (red). Scale bars: 250 µm. Measured *m/z* values and errors: (**A**) 579.534705 (+0.03 ppm); (**B**) 494.266309 (+0.05 ppm); (**C**) 760.584937 (−0.19 ppm); (**D**) 579.534705 (+0.03 ppm), 494.266309 (+0.05 ppm), 760.584937 (−0.19 ppm); (**E**) 480.250609 (−0.05 ppm), 494.266309 (+0.05 ppm). Adapted from [68] under the CC BY license 4.0.

**Figure 5 molecules-27-01414-f005:**
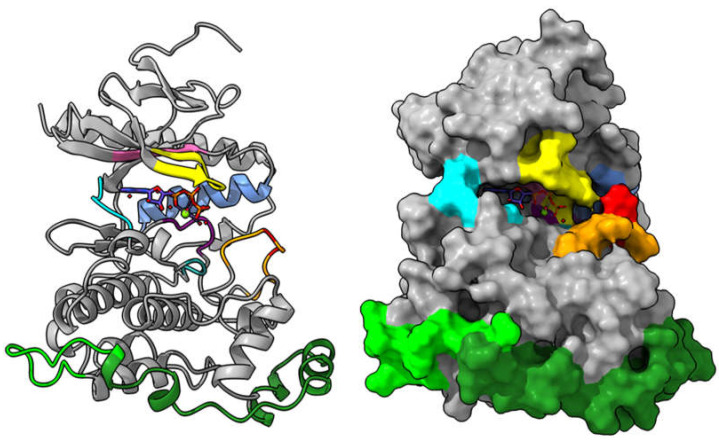
Structural model of the ATP-bound form of the *Schistosoma mansoni* JNK protein (UniProtKB acc. #A0A3Q0KT26) shown as ribbon (**left**) and surface representation (**right**). The protein was modeled utilizing the Steinegger advanced AlphaFold2 Google-Colab notebook [72] with custom settings. Structural elements were identified on the basis of the crystal structures of human JNK2 (PDB: 3e7o) and JNK3 (PDB: 1jnk) and are highlighted as follows: phosphorylation lip (orange) harboring the TxY motif (red), G-loop (yellow), C-helix (cornflower blue), hinge region (cyan), DFG-loop (purple), HRD (light sea green), AxKxL motif (hot pink) and the MAP-kinase insert (forest green) including the JNK insert (lime green). The most probable binding mode of ATP including two magnesium ions and 7 surrounding water molecules was derived from constructing an overlay of the *S. mansoni* AlphaFold2 structure model to the crystal structure of human JNK3 in complex with the ATP-analogue adenylyl imidodiphosphate (PDB: 1jnk).

**Table 1 molecules-27-01414-t001:** Comparison between traditional de novo drug discovery and drug repurposing.

De Novo Drug Discovery and Development	Drug Repurposing
Higher investment cost and financial return	Lower research, development (R&D) costs and more favorable financial return on investment
Long development time for new drugs	Reduces the drug development timeline (might not require Phase 1 clinical trials)
Ever-expanding toolbox of small-molecule compounds	Potential for reuse of compounds (despite evidence of adverse effects and failed efficacy in some indications)
Higher risk of failure	Higher probability of success
Focused on the discovery of drugs to treat chronic diseases and complex syndromes	Development of drugs for emerging and re-emerging infectious diseases

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
