# Peer review of "Drug Repurposing and De Novo Drug Discovery of Protein Kinase Inhibitors as New Drugs against Schistosomiasis"

_molecules, 2022, doi:10.3390/molecules27041414_

Round 1

Reviewer 1 Report

  1. Authors need to draw the structure of all the drug that were used for repuroposing studies.
  2. Need to screen some more similiar compounds or drugs with Protein Kinase Inhibitors against Schistosomiasis
  3. Grammatical and english errors need to be checked.

Author Response

A detailed rebuttal (with new Figures) has been uploaded as pdf

Reviewer 2 Report

The paper entitled “Drug Repurposing and de novo Drug Discovery of Protein Kinase Inhibitors as New Drugs against Schistosomiasis” is a review, which describes the protein kinase inhibitors against Schistosomiasis by using drug repurposing and de novo approach. Although the authors represented this work in a good format and accurate way, the manuscript is very concise. The review articles with a critical assessment of their topic raised by authors are very important as they can make easier the work of different research groups and provide new directions for all segments of research. In this review article, the authors mentioned (in the abstract) that they assessed critically the various approaches including screening of natural compounds, de-novo drug development, and drug-repurposing methods against schistosomiasis. However, their argument remains unclear and there is a lack of a critical assessment of the data. They do not represent a detailed study of natural compounds (except curcumin, quercetin, and genistein) and their cross-talk as an assessment that how to mitigate the protein kinase inhibition along with the anti-schistosomiasis activity of numerous natural compounds. A critical assessment data of these compounds are lacking here. Along with this, synthetic compounds were also represented poorly in the manuscript.  Detailed techniques and their cross-talk against drug repurposing and de-novo drug development programs are still missing. These are simply not addressed adequately.

In my opinion, this manuscript does not meet the requirements of the journal in this form. Therefore, I do not recommend this article for publication in the “Molecules” journal.

Author Response

(The authors gave the same response as above.)

Reviewer 3 Report

The present manuscript describes broad and crucial line approaches about new drugs to the treatment of the discovery and use of neglected diseases (schistosomiasis). The text show a great and consice discussion with a good English language grammar.
The study and investment in the reuse of pre-existing drugs has, in fact, many advantages in relation to investment in the development of new drugs for schistosomiasis.
The quality of the figures is excellent.

Line 76: (Table 1) "this is not necessary"

Author Response

(The authors gave the same response as above.)

Round 2

Reviewer 1 Report

The structure in the figure 1 and 2 is just copy and paste. The structure need to be drawn using Chemdraw.

Reviewer 2 Report

I recommend this manuscript for publication in the "Molecules" journal